# Cell-type heterogeneity in the early zebrafish olfactory epithelium is generated from progenitors within preplacodal ectoderm

Raphaël Aguillon[1†], Julie Batut[1†], Arul Subramanian[2], Romain Madelaine[1‡], Pascale Dufourcq[1], Thomas F Schilling[2], Patrick Blader[1]*

[1]Centre de Biologie du Développement (CBD, UMR5547), Centre de Biologie Intégrative (CBI, FR3743), Université de Toulouse, Toulouse, France; [2]Department of Developmental and Cell Biology, University of California, Irvine, United States

**Abstract** The zebrafish olfactory epithelium comprises a variety of neuronal populations, which are thought to have distinct embryonic origins. For instance, while ciliated sensory neurons arise from preplacodal ectoderm (PPE), previous lineage tracing studies suggest that both Gonadotropin releasing hormone 3 (Gnrh3) and microvillous sensory neurons derive from cranial neural crest (CNC). We find that the expression of Islet1/2 is restricted to Gnrh3 neurons associated with the olfactory epithelium. Unexpectedly, however, we find no change in Islet1/2+ cell numbers in *sox10* mutant embryos, calling into question their CNC origin. Lineage reconstruction based on backtracking in time-lapse confocal datasets, and confirmed by photoconversion experiments, reveals that Gnrh3 neurons derive from the anterior PPE. Similarly, all of the microvillous sensory neurons we have traced arise from preplacodal progenitors. Our results suggest that rather than originating from separate ectodermal populations, cell-type heterogeneity is generated from overlapping pools of progenitors within the preplacodal ectoderm.
DOI: https://doi.org/10.7554/eLife.32041.001

*For correspondence:
patrick.blader@univ-tlse3.fr

†These authors contributed equally to this work

Present address: ‡Beckman Center, Stanford University School of Medicine, Stanford, United States

Competing interests: The authors declare that no competing interests exist.

## Introduction

A fundamental question in developmental neurobiology is how different neuronal subtypes arise from fields of pluripotent progenitors. At the end of gastrulation, the anterior neural plate border of the zebrafish embryo gives rise to two specialized regions of ectoderm: the preplacodal ectoderm (PPE) that will ultimately produce the cranial placodes, and the cranial neural crest (CNC). Specification of the PPE is achieved through the action of so-called preplacodal competence factors such as *tfap2a*, *tfap2c*, *foxi1* and *gata3* (*Kwon et al., 2010*). During a similar time-window, key neural crest specifier genes, such as *foxd3* (*Lister et al., 2006*; *Montero-Balaguer et al., 2006*; *Stewart et al., 2006*), *tfap2a* (*Barrallo-Gimeno et al., 2004*) and *sox10* (*Dutton et al., 2001b*) establish the CNC fate. Cranial placodes subsequently arise via the condensation of specific regions within the PPE along the anteroposterior axis, with the adenohypophyseal and olfactory placodes forming anteriorly, the lens and trigeminal placodes forming at an intermediate position and the otic, lateral line and epibranchial placodes forming posteriorly (for review see [*Aguillon et al., 2016*]). Concomitantly, CNC cells delaminate and migrate throughout the head, where they have been reported to contribute to a large number of cell types, including sensory and neurosecretory cells associated with the olfactory system (*Whitlock et al., 2003*; *Saxena et al., 2013*). This dual embryonic (PPE/CNC) origin for olfactory neurons in zebrafish may have critical developmental and functional consequences.

In zebrafish embryos, olfactory neurons are generated in two waves, early olfactory neurons (EON) and olfactory sensory neurons (OSN), under the redundant control of the bHLH proneural transcriptions factors Neurog1 and Neurod4 (*Madelaine et al., 2011*). EONs act as pioneers for the establishment of projections from the olfactory epithelium to the olfactory bulb. Once OSN projections are established, a subset of EONs dies by apoptosis (*Whitlock and Westerfield, 1998*). This suggests the existence of distinct subtypes of neurons within the EON population, but specific markers for these different subtypes have yet to be described. Neural subtype heterogeneity is also detected early within the OSN population; in zebrafish the predominant subtypes are ciliated sensory neurons that have long dendrites and express olfactory marker protein (OMP) and microvillous sensory neurons, which have short dendrites and express the Transient receptor potential cation channel, subfamily C, member 2b (Trpc2b)(*Hansen and Zeiske, 1998*; *Sato et al., 2005*). A third neural subtype associated with the early olfactory epithelium in zebrafish expresses *gonadotropin releasing hormone 3* (*gnrh3*). Rather than projecting exclusively to the olfactory bulb, Gnrh3 neurons send their axons caudally to various brain regions, including the hypothalamus (*Abraham et al., 2008*). Like GnRH1 neurons in amniotes, zebrafish Gnrh3 neurons leave the proximity of the olfactory epithelium and can be found along a migratory tract from the terminal nerve to the ventral telencephalon at late larval stages. While laser-ablating Gnrh3+ cells leads to sterility, animals homozygous for TALEN-induced mutations of *gnrh3* are fertile, pointing to the need for identifying other genes expressed in these cells that might underlie the differences between these phenotypes (*Abraham et al., 2010*; *Spicer et al., 2016*).

Although the major neural cell types associated with the olfactory epithelium appear to be conserved across vertebrates, there is no coherent vision as to their lineage origin between species. For instance, while Gnrh cells associated with the developing olfactory epithelium are reported to be of preplacodal origin in chick, in the zebrafish they have been shown to derive from the neural crest (*Whitlock et al., 2003*; *Sabado et al., 2012*); in mouse, Cre/*lox* experiments suggest that Gnrh cells are of mixed lineage origin, coming from both the ectoderm and CNC (*Forni et al., 2011*). To identify additional markers of cell-type heterogeneity in the developing zebrafish olfactory epithelium we screened expression of molecules known to label discrete sets of neurons in other regions of the nervous system. We found that an antibody that recognizes the Islet family (Islet1/2) of LIM-homeoproteins labels Gnrh3 neurons in the olfactory epithelium (*Ericson et al., 1992*). We find no change in the numbers of Islet1/2+ cells in the olfactory epithelium in *sox10* mutant embryos, which are deficient in many CNC lineages. This is in contrast with previous studies and calls into question the proposed CNC origin of Gnrh+ cells. Consistent with these findings, lineage reconstructions of time-lapse confocal movies show that most if not all Gnrh3+ neurons, as well as microvillous sensory neurons, derive from the PPE. Thus, cell-type heterogeneity within the olfactory epithelium is likely established entirely from progenitors within the PPE.

## Results

### Islet1/2 expression in Gnrh3 neurons in the olfactory epithelium is unaffected in *sox10* mutants

Heterogeneity in neuronal subtypes is apparent in the zebrafish olfactory epithelium from early developmental stages (*Whitlock and Westerfield, 1998*, *2000*; *Whitlock et al., 2003*; *Sato et al., 2005*; *Madelaine et al., 2011*; *Saxena et al., 2013*). While searching for novel markers of this heterogeneity, we found that at 48 hr post-fertilization (hpf) immunoreactivity to the Islet1/2 monoclonal antibody 39.4D5 is restricted to a small group of cells in the olfactory epithelium at the interface with the telencephalon (*Figure 1A*). The number and position of these Islet1/2+ cells resembles expression of *gonadotropin releasing hormone 3* (*gnrh3*) (*Gopinath et al., 2004*). To determine if they are the same cells, we examined expression of Islet1/2 in a transgenic line expressing enhanced GFP (eGFP) under the control of the *gnrh3* promoter, which recapitulates the endogenous expression of *gnrh3* associated with the olfactory epithelium (*Abraham et al., 2008*). Our results reveal a complete overlap between eGFP expression from the *Tg(gnrh3:eGFP)* transgene and Islet1/2 (*Figure 1A*).

It has been reported that zebrafish Gnrh3+ neurons associated with the olfactory epithelium derive from cranial neural crest (CNC), and that impairing CNC specification through morpholino

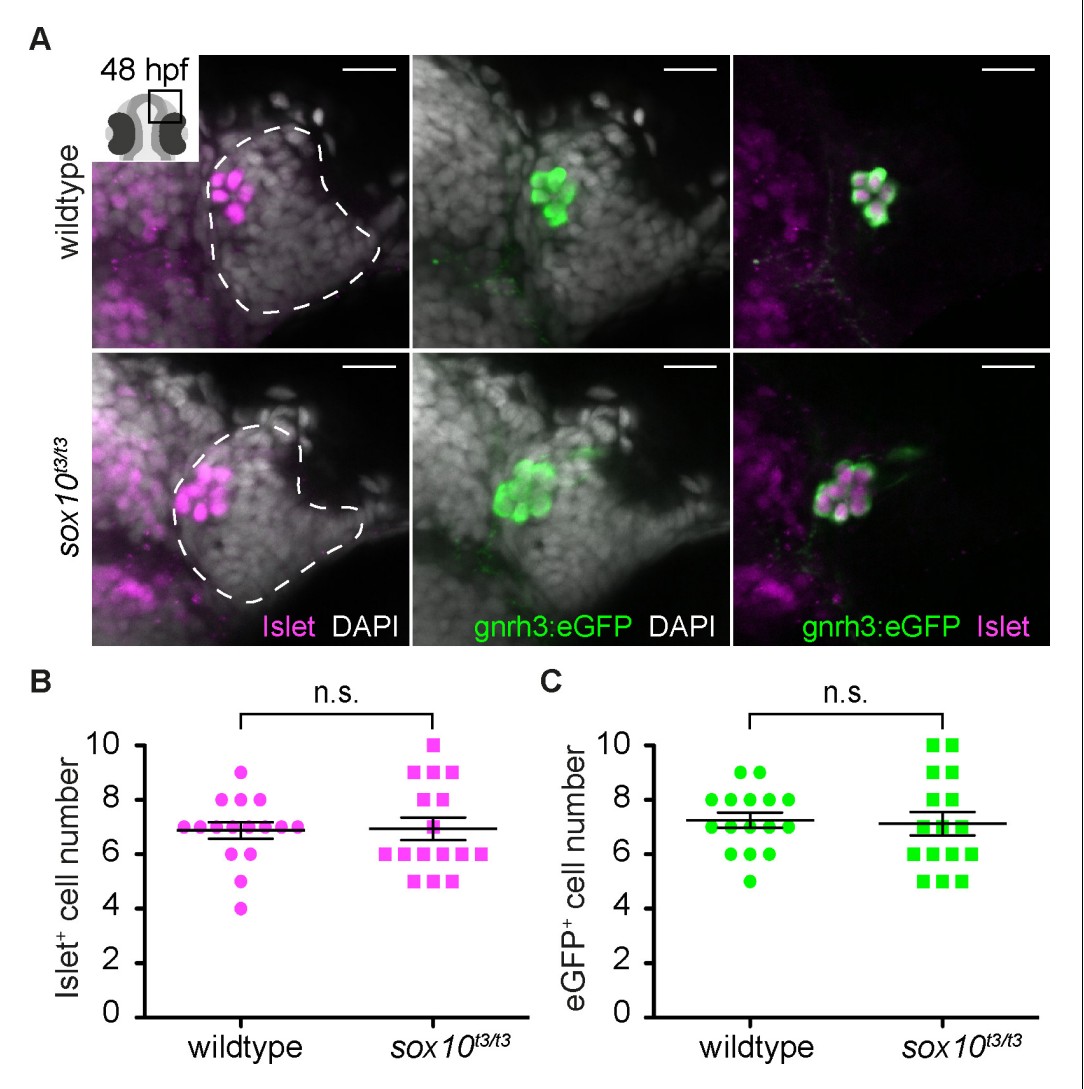

**Figure 1.** Islet1/2 labels Gnrh3 neurons of the olfactory epithelium and its expression is unaffected in *sox10* mutants. (**A**) Single confocal sections of olfactory epithelia from *Tg(gnrh3:eGFP)* embryos at 48 hpf, in either wildtype or *sox10^{t3/t3}* mutants, labelled for the expression of Islet1/2 (magenta) and eGFP (green); nuclei are labelled with DAPI (grey). Embryos are viewed dorsally and the inset in the first panel shows a schematic representation of a 48 hpf embryonic head indicating the region analysed. Dotted lines highlight the olfactory epithelia and scalebars represent 15 μm. (**B**) Counts of cells expressing Islet1/2 in either wildtype or *sox10^{t3/t3}* mutant epithelia (6.8 ± 0.3 versus 6.9 ± 0.4 cells per epithelium, n = 16 epithelia). (**C**) Counts of cells expressing eGFP from the *Tg(gnrh3:eGFP)* transgene in either wildtype or *sox10^{t3/t3}* mutant placodes (7.2 ± 0.3 versus 7.1 ± 0.4 cells per epithelium, n = 16 epithelia). (**B–C**) Mean ±s.e.m. P values are calculated using a two-tailed Student's t-test, n.s. not significant.

DOI: https://doi.org/10.7554/eLife.32041.002

The following source data and figure supplement are available for figure 1:

**Source data 1.** Islet+ cell number quantification.
DOI: https://doi.org/10.7554/eLife.32041.004

**Source data 2.** eGFP+ cell number quantification.
DOI: https://doi.org/10.7554/eLife.32041.005

**Figure supplement 1.** Expression of endogenous *gnrh3* in *sox10^{t3/t3}* mutants and wildtype siblings.
DOI: https://doi.org/10.7554/eLife.32041.003

knock-down of the HMG transcription factor *sox10* dramatically reduces their number (*Whitlock et al., 2003*, *2005*). Unexpectedly, we found no difference in Islet1/2+ cell numbers in embryos homozygous for a strong *sox10* mutant allele ($sox10^{t3/t3}$) relative to wildtype siblings ($6.9 \pm 0.4$ versus $6.8 \pm 0.3$ cells per epithelium, n = 16 epithelia from 9 wildtype and 16 epithelia from 8 mutant embryos; *Figure 1A and B*; [*Dutton et al., 2001b*]). Similarly, eGFP expression in embryos carrying the *Tg(gnrh3:eGFP)* transgene was unaffected in a $sox10^{t3/t3}$ mutant background ($7.1 \pm 0.4$ versus $7.2 \pm 0.3$ cells per epithelium, n = 16 epithelia from 9 wildtype and 16 epithelia from 8 mutant embryos; *Figure 1A,C*); endogenous *gnrh3* expression was also unaffected in *sox10* mutant embryos, ruling out a transgene-specific effect (*Figure 1—figure supplement 1*). Our results indicate that Islet1/2 is a novel marker of Gnrh3 neurons associated with the zebrafish olfactory epithelium, and that these cells do not require Sox10 for specification.

A previous study assessed the origin of zebrafish Gnrh3 neurons by DiI labelling of premigratory CNC followed by immunostaining for GnRH (*Whitlock et al., 2003*). To revisit the proposed CNC origin of these neurons, we chose a Cre/*lox*-based approach coupled with analysis of Islet1/2 expression. Double-heterozygous embryos carrying both a *Tg(−28.5Sox10:Cre)* and *Tg(ef1a:loxP-DsRed-loxP-eGFP)* transgene display a permanent shift from DsRed to eGFP expression in CNC lineages (*Kague et al., 2012*). Despite widespread expression of eGFP throughout the heads of 48 hpf double-heterozygous *Tg(−28.5Sox10:Cre);Tg(ef1a:loxP-DsRed-loxP-egfp)* embryos, including in cells surrounding the olfactory epithelium, we did not detect eGFP;Islet1/2 double-positive cells in the epithelia themselves (n = 20 epithelia from 10 embryos; *Figure 2* and *Figure 2—figure supplement 1*). While these are negative results, when combined with the lack of defects in Islet1/2+ cell numbers in *sox10* mutant embryos they lend further support against a CNC origin for Gnrh3 neurons in the developing zebrafish olfactory system.

## Lineage reconstruction reveals an anterior preplacodal ectoderm origin for Gnrh3 neurons

Gnrh3+ neurons associate closely with the olfactory epithelium from early stages (*Gopinath et al., 2004*). However, a lack of Cre lines specific for progenitors of the olfactory epithelium precluded using a Cre/*lox* approach to address if Gnrh3 neurons derive from these cells. As an alternative, we developed an unbiased backtracking approach using time-lapse confocal movies. Briefly, synthetic mRNAs encoding Histone2B-RFP (H2B-RFP) were injected into *Tg(gnrh3:eGFP)* transgenic embryos, which were subsequently imaged from 12 to 36 hpf (*Figure 3A*); delamination and migration of CNC begins approximately 2 hr after the initiation of the time-lapse acquisition, and eGFP from the

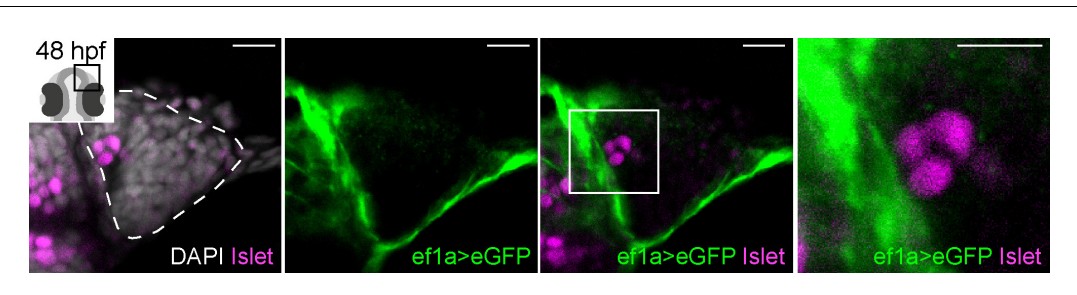

**Figure 2.** Cre/*lox* lineage mapping of neural crest cells reveals contributions to cells surrounding the olfactory epithelium but not to Islet/Gnrh3 + neurons. Single confocal sections of an olfactory epithelium from a double-heterozygous *Tg(−28.5Sox10:Cre);Tg(ef1a:loxP-DsRed-loxP-egfp)* embryo at 48 hpf, labelled for the expression of Islet1/2 (magenta) and eGFP (green); nuclei are labelled with DAPI (grey). No eGFP;Islet1/2 double-positive cells are detected in the epithelia. Embryos are viewed dorsally and the inset in the first panel shows a schematic representation of a 48 hpf embryonic head indicating the orientation of the region analysed. The dotted line highlights the olfactory epithelium and scalebars represent 15 µm in the low and 10 µm in the high magnifications, respectively; the region shown in the high magnification is indicated in the neighbouring panel.
DOI: https://doi.org/10.7554/eLife.32041.006

The following figure supplement is available for figure 2:

**Figure supplement 1.** Expression of eGFP is detected throughout the head after switching of the *loxP-DsRed-loxP-egfp* cassette, including in known neural crest-derived lineages.
DOI: https://doi.org/10.7554/eLife.32041.007

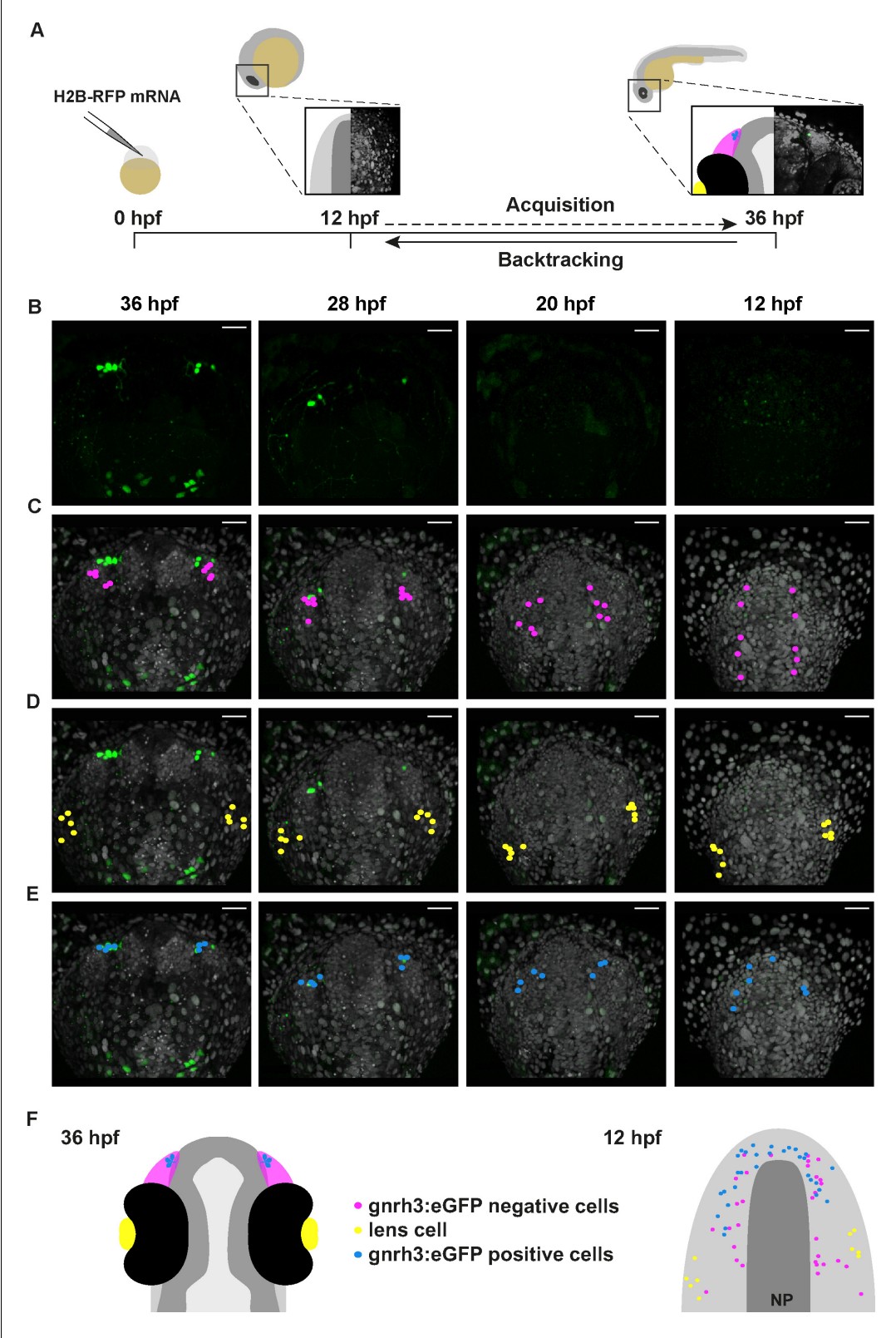

**Figure 3.** Lineage reconstruction reveals an anterior preplacodal ectoderm origin for Gnrh3 neurons: backtracking. (**A**) Schematic representation of the backtracking strategy. Synthetic mRNAs encoding Histone2B-RFP (H2B-RFP) were injected into *Tg(gnrh3:eGFP)* transgenic embryos, which were subsequently imaged from 12 to 36 hpf. The lineages of various populations of cells were manually retraced by backtracking H2B-RFP+ nuclei to their position at the beginning of the time-lapse series. (**B–E**) Confocal projections from a representative 4D dataset at 36, 28, 20 and 12 hpf showing the
*Figure 3 continued on next page*

*Figure 3 continued*

GFP channel alone (**B**), the position of the nuclei backtracked from 10 gnrh3:eGFP-negative (**C**, pink), 10 lens (**D**, yellow) and 7 gnrh3:eGFP-positive (**E**, blue) cells at each timepoint. The embryo is shown with anterior up; GFP expression detected caudally is ectopic and does not reflect endogenous *gnrh3* expression. Scalebars represent 40 μm. (**F**) Schematic representation of an embryonic head at 36 hpf and the anterior neural plate (NP, dark grey) and adjacent preplacodal ectoderm (light grey) at 12 hpf. The 36 hpf head shows the colour code of the backtracked lineages, and the position of backtracked nuclei at 12 hpf is indicated in the preplacodal ectoderm. The 12 hpf representation shows the results obtained from 9 epithelia (5 embryos) for 30 gnrh3:eGFP-positive cells and 32 gnrh3:eGFP-negative cells; the 10 lens cells were backtracked from a pair of epithelia in a single 4D dataset only.

DOI: https://doi.org/10.7554/eLife.32041.008

The following figure supplement is available for figure 3:

**Figure supplement 1.** Backtracking data from individual *Tg(gnrh3:eGFP)* embryos.

DOI: https://doi.org/10.7554/eLife.32041.009

transgene is robustly expressed at 36 hpf (*Schilling and Kimmel, 1994*; *Abraham et al., 2008*). The lineage of various populations of cells was then manually retraced by backtracking H2B-RFP+ nuclei to their position at the beginning of the time-lapse series using Imaris software (*Figure 3A–E* and *Video 1*). To test our approach relative to well-established fate maps already generated for zebrafish cranial placode derivatives, we first backtracked H2B-RFP+ nuclei of gnrh3:eGFP-negative cells in the olfactory epithelia as well as lens cells, which can be identified at the end of the time-lapse series by their distinct morphology (*Figure 3C,D*). Consistent with previous lineage studies (*Whitlock and Westerfield, 2000*), we found that gnrh3:eGFP-negative cells of the olfactory epithelia derived from progenitors in the preplacodal ectoderm (PPE) at the anterior/lateral neural plate border (n = 32 cells from 9 epithelia of 5 embryos; *Figure 3C,F*, *Video 2* and *Figure 3—figure supplement 1*). Also as expected from earlier studies (*Dutta et al., 2005*), the H2B-RFP+ nuclei of lens cells traced back to a PPE domain caudal and slightly lateral to that of progenitors of the olfactory epithelia (n = 10 cells from 2 lenses of 1 embryo; *Figure 3D,F* and *Video 2*). In contrast to previous reports, the nuclei of gnrh3:eGFP-positive cells traced back to the most anterior region of the PPE (n = 30 cells from 9 epithelia of 5 embryos; *Figure 3B,E,F*, *Video 2* and *Figure 3—figure supplement 1*), rather than from a region

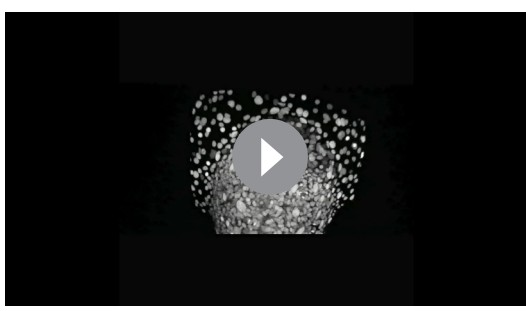

**Video 1.** Lineage reconstruction reveals an anterior preplacodal ectoderm origin for Gnrh3 neurons: backtracking. Movie showing an acquisition series and backtracking of a Gnrh3 neuron in a Histone2B-RFP loaded *Tg(gnrh3:eGFP)* transgenic embryo. The movie is divided into 5 parts: an acquisition phase, a phase showing a z-stack of the olfactory epithelium at 36 hpf, an initial backtracking phase, the unique mitosis detected during the backtracking, and a final backtracking phase. During backtracking, the nucleus being followed is labelled with a blue dot. During the mitosis, the sister cell is labelled with a pink dot for three frames until the two sister nuclei fuse. The movie finishes with a schematic representation of the anterior neural plate and adjacent preplacodal ectoderm at 12 hpf showing the results obtained from backtracking 30 gnrh3:eGFP-positive cells. The cell backtracked in the movie is indicated (arrowhead).

DOI: https://doi.org/10.7554/eLife.32041.010

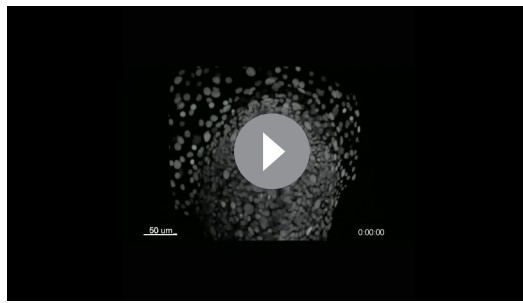

**Video 2.** Lineage reconstruction reveals an anterior preplacodal ectoderm origin for Gnrh3 neurons: tracks. Movie showing an acquisition series and backtracking of gnrh3:eGFP-negative olfactory cells, lens cells and gnrh3:eGFP-positive olfactory cells in an Histone2B-RFP loaded *Tg(gnrh3:eGFP)* transgenic embryo. The movie is divided into 5 parts: an acquisition phase, a backtracking phase for each lineage with tracks and a final representation of the anterior neural plate and adjacent preplacodal ectoderm at 12 hpf showing the combined tracks obtained for the three lineages.

DOI: https://doi.org/10.7554/eLife.32041.011

caudal to the lens progenitors as would be expected for CNC cells.

To confirm the anterior PPE origin of Gnrh3 neurons, we used photoconversion. We loaded *Tg (gnrh3:eGFP)* embryos with NLS-mEos2 by mRNA injection and at 12 hpf photoconverted either the left or right half of the domain predicted by backtracking to contain precursors of Gnrh3 neurons at this stage. Photoconverted embryos were then allowed to develop until 36 hpf, at which stage we determined if any gnrh3:eGFP+ cells had photoconverted nuclei (NLS-mEos2$^{PC}$; *Figure 4A–D*). Photoconversion caused no changes in the number of gnrh3:eGFP+ cells ($5.4 \pm 0.5$ versus $5.2 \pm 0.6$ cells per epithelium, n = 9 control and 9 converted epithelia from 9 embryos; *Figure 4E*). However, while on average less than one NLS-mEos2$^{PC}$;gnrh3:eGFP+ double labeled cell per epithelium was detected on the control side of the embryo ($0.3 \pm 0.2$ cells per epithelium, n = 9 epithelia from 9 embryos), the entire gnrh3:eGFP+ population was double-labeled on the photoconverted side ($4.6 \pm 0.3$ cells per epithelium, n = 9 epithelia from 9 embryos; *Figure 4E*). There was no statistical difference between the number of gnrh3:eGFP+ and eGFP/mEos2$^{PC}$-double-positive cells on the photoconverted side suggesting that all Gnrh3 neurons associated with the olfactory epithelium are derived from the anterior PPE.

## Microvillous sensory neurons also derive from the preplacodal ectoderm

Zebrafish microvillous sensory neurons, like their counterparts in the rodent vomeronasal organ, express Transient receptor potential cation channel, subfamily C, member 2 (Trpc2) and VR-type olfactory receptors (*Sato et al., 2005*). The expression of a *Tg(−4.5trpc2b:GAP-Venus)* transgene is reduced in *sox10* morphant embryos suggesting a CNC origin for microvillous sensory neurons in zebrafish (*Saxena et al., 2013*), as initially thought for Gnrh3+ neurons. Given our results with Gnrh3 neurons in *sox10* mutant embryos (*Figure 1A,B*; [*Whitlock et al., 2005*]), we also revisited the microvillous lineage. Indeed, we found that the expression of endogenous *trpc2b* was unaffected in *sox10* mutant embryos (*Figure 5—figure supplement 1*). We then backtracked eGFP+ neurons in the *Tg (−4.9sox10:eGFP)* transgenic background (*Video 3*; [*Wada et al., 2005*]); expression from this transgene has been shown to overlap almost completely with that of *Tg(−4.5trpc2b:GAP-Venus)* (*Saxena et al., 2013*). As before, nuclei of randomly chosen eGFP-negative cells in the olfactory epithelium trace back to a position in the PPE at the anterior/lateral neural plate border (n = 44 cells from 7 epithelia of 4 embryos; *Figure 5A,D*, *Video 4* and *Figure 5—figure supplement 2*); backtracked lens cells also behaved as before (n = 10 cells from 2 lenses of 1 embryo; *Figure 5B,D* and *Video 4*). Unexpectedly, backtracking revealed a similar PPE origin for sox10:eGFP+ cells as their eGFP-negative neighbours (n = 41 cells from 7 epithelia of 4 embryos; *Figure 5C,D*, *Video 4* and *Figure 5—figure supplement 2*).

Similar to the Gnrh3+ cells, we used photoconversion of NLS-mEos2 to confirm our backtracking results for the microvillous population. This time, however, photoconversion was focused on the left or right two-thirds of the anterior PPE of *Tg(−4.9sox10:eGFP)* embryos (*Figure 6A,B*). As before, the number of eGFP+ cells was unaffected by the photoconversion ($8 \pm 0.6$ versus $9.8 \pm 0.7$ cells per epithelium, n = 9 converted and 6 non-converted epithelia from 9 embryos; *Figure 6C–E*), and less than one NLS-mEos2$^{PC}$;sox10:eGFP+ cell per epithelium was detected on the control side of the embryo ($0.3 \pm 0.3$ cells per epithelium, n = 6 epithelia from 6 embryos; *Figure 6E*). On the photoconverted side, however, an average of just under 60% of the total sox10:eGFP+ population was also NLS-mEos2$^{PC}$-positive ($4.6 \pm 0.6$ cells per epithelium, n = 9 epithelia from 9 embryos; *Figure 6D,E*). This is consistent with the fact that we photoconverted approximately two-thirds of the cells in the PPE that gives rise to the olfactory epithelium.

Taken together, our results obtained using a combination of backtracking and photoconversion indicate that Gnrh3 and microvillous sensory neurons associated with the zebrafish olfactory epithelium derive from the anterior PPE.

## Discussion

As part of our ongoing studies into the mechanisms underlying neurogenesis in the early zebrafish olfactory epithelium (*Madelaine et al., 2011*), here we have investigated how different subtypes of olfactory neurons arise. Using a novel marker for Gnrh3 neurons we have revisited this lineage and its origins within the epithelium. By combining Cre/*lox*, backtracking in 4D confocal datasets and

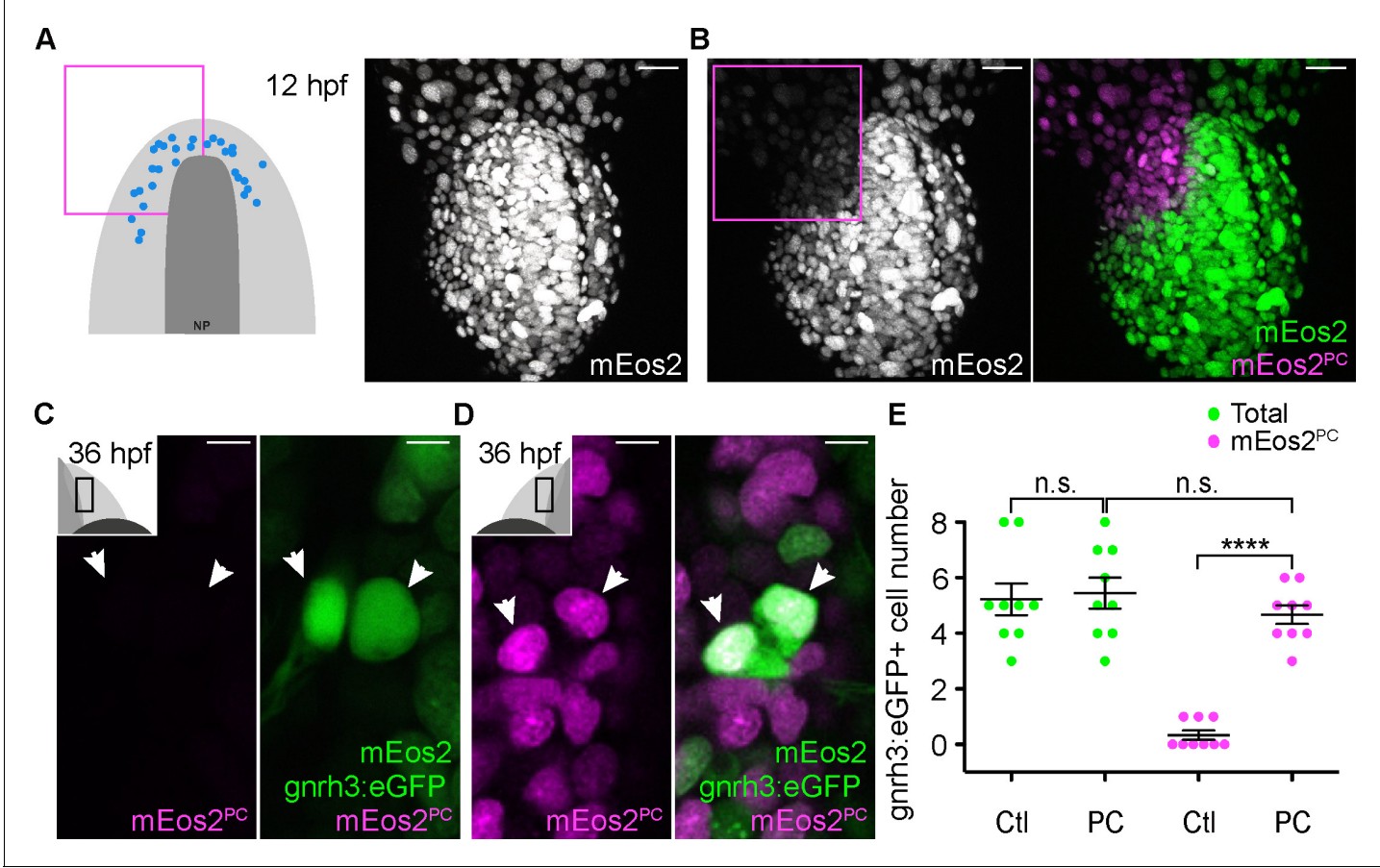

**Figure 4.** Lineage reconstruction reveals an anterior preplacodal ectoderm origin for Gnrh3 neurons: photoconversion. (A,B) Confocal projections of NLS-mEos2 loaded, *Tg(gnrh3:eGFP)* embryos at 12 hpf before (A) and after photoconversion (B). The first panel in (A) shows a schematic representation of the anterior neural plate (NP, dark grey) and adjacent preplacodal ectoderm (light grey) at 12 hpf indicating the origin of backtracked gnrh3:eGFP-positive cells (blue dots) and the photoconverted area (magenta square), which is also indicated on the confocal projection shown in (B) after photoconversion. Scalebars represent 30 µm. (C,D) Single confocal sections of olfactory epithelia from *Tg(gnrh3:eGFP)* embryos at 36 hpf showing the expression of eGFP from the transgene (cytoplasmic green), unconverted NLS-mEos2 (mEos2; nuclear green) and converted NLS-mEos2 (mEos2$^{PC}$; nuclear magenta). Insets in (C) and (D) shows a schematic representation of an embryonic head at 36 hpf indicating the region analysed (black rectangle). Scalebars represent 5 µm. (E) Counts of cells expressing eGFP from the *Tg(gnrh3:eGFP)* transgene or eGFP and photoconverted mEos2 (mEos2$^{PC}$) on the control (Ctl) versus photoconverted (PC) sides of the embryo at 36 hpf. No difference in the number of eGFP-positive cells is apparent between the Ctl and PC conditions (5.2 ± 0.6 versus 5.4 ± 0.5 cells per epithelium, n = 9 epithelia). Conversely, while a number of cells equivalent to the entire gnrh3:eGFP+ population is also mEos2$^{PC}$-positive on the photoconverted side, virtually no eGFP/mEos2$^{PC}$-double positive cells are detected on the control side (4.6 ± 0.3 versus 0.3 ± 0.2 cells per epithelium, n = 9 epithelia). Mean ±s.e.m. P values are calculated using a two-tailed Student's t-test, n.s. not significant, ****p<0.0001.

DOI: https://doi.org/10.7554/eLife.32041.012

The following source data is available for figure 4:

**Source data 1.** gnrh3:eGFP+ and mEos2$^{PC}$+ cell number quantification.
DOI: https://doi.org/10.7554/eLife.32041.013

photoconversion, we show that Gnrh3+ neurons derive from progenitors in the PPE; similar live imaging techniques also indicate a PPE origin for microvillous sensory neurons. These results support a common PPE origin for all of the neuronal populations within the olfactory epithelium and argue against any CNC contribution. More generally, they suggest a mechanism by which cellular heterogeneity arises progressively within a field of neuronal progenitors.

In zebrafish at least two Islet genes, *islet1* and *islet2b*, are expressed in the olfactory Gnrh3+ cells (R. Aguillon and P. Blader, unpublished data). Whether or not Islet transcription factors regulate the development of Gnrh3 neurons, similar to Islet1 in pancreatic β-cells, remains unknown

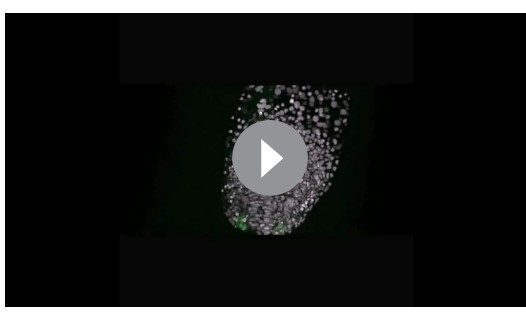

**Video 3.** Microvillous sensory neurons derive from classical preplacodal precursors: backtracking. Movie showing an acquisition series and backtracking of a microvillous sensory neuron in a Histone2B-RFP loaded *Tg(sox10:eGFP)* transgenic embryo. The movie is divided into 5 parts: an acquisition phase, a phase showing a z-stack of the olfactory epithelia at 36 hpf, an initial backtracking phase, the unique mitosis detected during the backtracking, and a final backtracking phase. During backtracking, the nucleus being followed is labelled with a blue dot. During the mitosis, the sister cell is labelled with a pink dot for three frames until the two sister nuclei fuse. The movie finishes with a schematic representation of the anterior neural plate, the adjacent preplacodal ectoderm (light grey) and neural crest (CNC) at 12 hpf showing the results obtained from backtracking 41 sox10:eGFP-positive cells located within the olfactory epithelium at 36 hpf. The cell backtracked in the movie is indicated (arrowhead).

DOI: https://doi.org/10.7554/eLife.32041.017

(*Ahlgren et al., 1997*). If this is the case, the specification of endocrine lineages, like Gnrh3-expressing olfactory cells and Insulin-expressing β-cells, may be an ancestral role of Islet proteins, which were later co-opted into other lineages such as motor neurons in the spinal cord (*Karlsson et al., 1990*; *Ericson et al., 1992*; *Appel et al., 1995*). Our analysis also reassigns the Gnrh3 lineage to progenitors found in a domain of the PPE that, in addition to producing olfactory lineages, is known to give rise to neurons of the adenohypophysis (*Dutta et al., 2005*) and proposed to generate hypothalamic Gnrh2 neurons (*Whitlock et al., 2003*). Whether or not this region of the zebrafish PPE provides an environment that promotes the production of neuro-endocrine lineages remains an open question.

Lineage reconstruction experiments in the tunicate, *Ciona intestinalis*, recently demonstrated the existence of a proto-placodal ectoderm, equivalent to the vertebrate PPE, which gives rise to both sensory neurons and Gnrh cells (*Abitua et al., 2015*). We propose that the development of a Gnrh population from the anterior PPE has been conserved during chordate evolution. Whether or not this conservation holds for other vertebrates, however, remains unclear. DiI labeling experiments in the chick suggest that GnRH cells derive from precursors of the olfactory epithelium separate from the adenohypophysis (*Sabado et al., 2012*), while our study and others show that in zebrafish these two populations overlap significantly in the PPE (reviewed in [*Toro and Varga, 2007*]). In mice Cre/*lox* lineage analysis suggests that GnRH cells associated with the olfactory epithelium are of mixed origin, being derived 70% from the ectoderm and 30% from CNC (*Forni et al., 2011*). This does not appear to be the case in chick, however, as grafted neural folds expressing GFP do not contribute to the olfactory epithelium (*Sabado et al., 2012*). Furthermore, similar to our results in zebrafish no change in GnRH cells numbers is detected in *Sox10*-null mutant mice (*Pingault et al., 2013*). As the original lineage assignment in mouse was established using one ectodermal (*Crect*; [*Reid et al., 2011*]) and one neural crest (*Wnt1Cre*; [*Danielian et al., 1998*]) Cre line, our results suggest that revisiting the lineage assignment in the mouse with other genetic tools or other approaches is needed.

The origin of zebrafish microvillous sensory neurons is controversial (*Saxena et al., 2013*; *Torres-Paz and Whitlock, 2014*). Our results indicate that this lineage is derived from progenitors in the PPE. Furthermore, they imply that the expression of eGFP in the olfactory epithelium of *Tg(sox10: eGFP)* embryos does not reflect a CNC origin for these cells but rather an ectopic site of transgene expression. Why ectopic expression of the transgene in the olfactory epithelium appears restricted to microvillous sensory neurons is unclear but highlights the need to be cautious when using transgenic tools in lineage analyses. In this regard, the backtracking approach we have developed provides a powerful alternative for ascertaining lineage assignments during zebrafish embryogenesis. While in this study we used transgenic lines to identify cell types for backtracking, with the caveats that this obliges, we have also backtracked cells identified using antibody markers. For this, cells for backtracking can be chosen by comparing the 3D architecture of a tissue described by nuclear position at the end of a time-lapse series to that of the same embryo after fixation and antibody labeling.

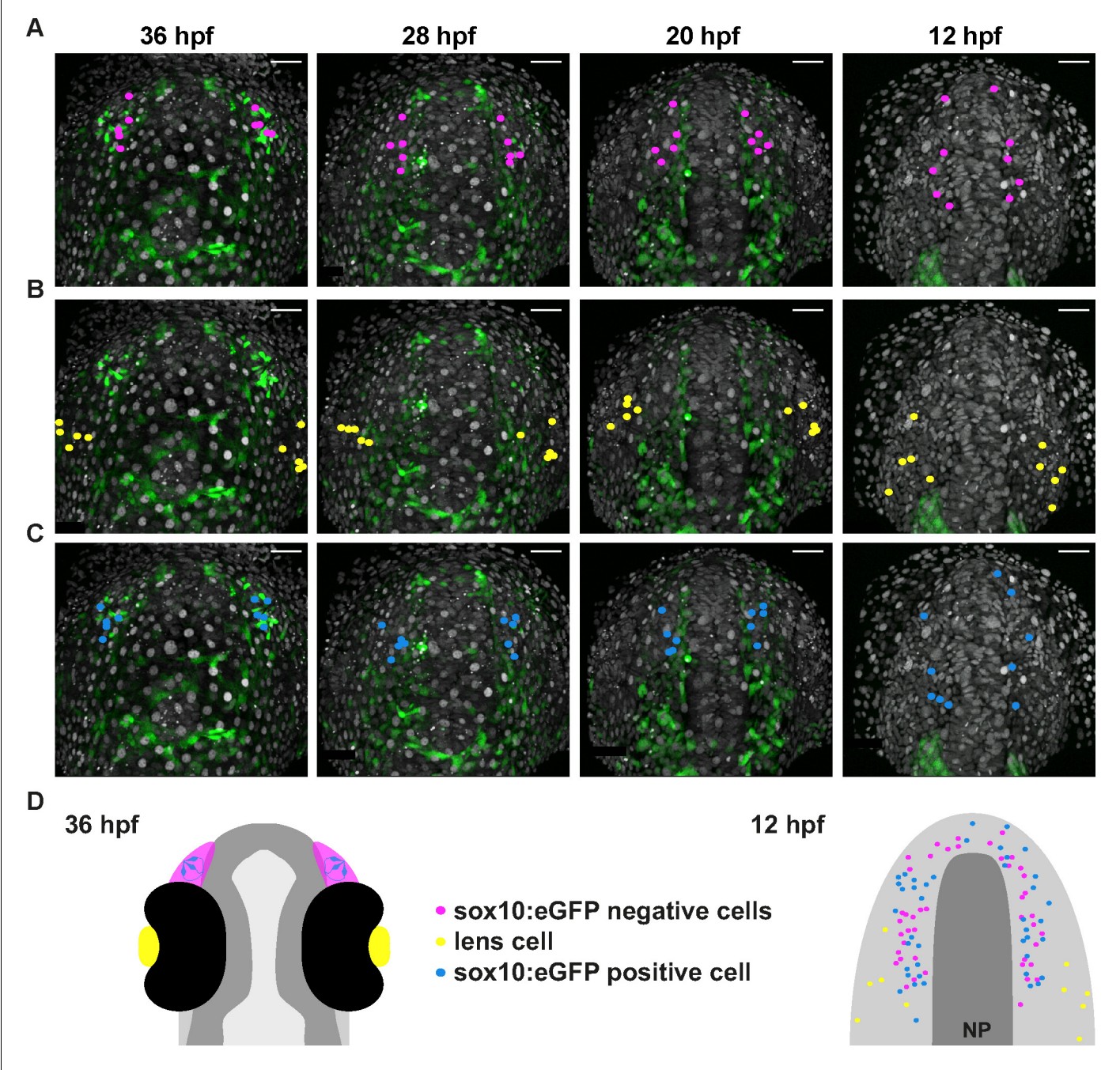

**Figure 5.** Microvillous sensory neurons derive from preplacodal precursors: backtracking. (A–C) Confocal projections from a representative 4D dataset at 36, 28, 20 and 12 hpf showing the position of the nuclei backtracked from 10 sox10:eGFP-negative (A, pink), 10 lens (B, yellow) and 10 sox10:eGFP-positive (C, blue) cells at each timepoint. The embryo is shown with anterior up. Scalebars represent 40 μm. (D) Schematic representation of an embryonic head at 36 hpf and the anterior neural plate (NP, dark grey) and adjacent preplacodal ectoderm (light grey) at 12 hpf; the cranial neural crest (CNC) at the earlier stage is indicated. While the 36 hpf head shows the colour code of the backtracked lineages, the position of backtracked nuclei at 12 hpf is indicated in the preplacodal ectoderm. The 12 hpf schema represents the results obtained from 7 epithelia (4 embryos) for 41 sox10:GFP-positive cells and 44 sox10:eGFP-negative cells; the 10 lens cells were backtracked from a pair of epithelia in a single 4D dataset only.

DOI: https://doi.org/10.7554/eLife.32041.014

The following figure supplements are available for figure 5:

**Figure supplement 1.** Expression of endogenous *trpc2b* is unchanged in *sox10^{t3/t3}* mutant embryos relative to wildtype siblings.
DOI: https://doi.org/10.7554/eLife.32041.015

**Figure supplement 2.** Backtracking data from individual *Tg(sox10:eGFP)* embryos.

*Figure 5 continued on next page*

*Figure 5 continued*

DOI: https://doi.org/10.7554/eLife.32041.016

Our analysis also highlights unexpected difficulties with using morpholinos. It has been shown that morpholinos can produce phenotypes that are not seen in genetic mutants, either due to non-specific p53-dependent effects or mechanisms that compensate for mutations (*Gerety and Wilkinson, 2011*; *Kok et al., 2015*; *Rossi et al., 2015*). Previous studies have shown a very good correlation between *sox10* mutant and morphant phenotypes in a variety of tissues, including neural crest derivatives like melanophores and dorsal root ganglia neurons and glia, as well as the ear (*Dutton et al., 2001a*, *2001b*; *Carney et al., 2006*); embryos homozygous for the *sox10^{t3}* allele used in this study displayed the previously reported strong pigment and ear phenotypes. Taken together, this supports the idea that non-specific effects in *sox10* morphants are limited to the olfactory system. Intriguingly, *foxd3* morphants also display a reduction in the number of Gnrh3 neurons associated with the olfactory epithelium (*Whitlock et al., 2005*). We have shown that the Gnrh3 lineage is PPE-derived and, therefore, the effect of these morpholinos cannot be explained by genetic compensation in a subset of the neural crest. The decrease in Gnrh3 cell numbers in *sox10* and *foxd3* morphants might reflect a tissue-specific p53-dependent effect. Alternatively, the morphant phenotype might indicate that morpholino-induced developmental delays are more pronounced in the olfactory system than elsewhere. Whatever the cause, our finding suggests that extra care should be taken when using morpholinos to study olfactory development.

In conclusion, our results argue that cell-type heterogeneity in the zebrafish olfactory epithelium is generated from progenitors within the PPE, and begin to provide coherence for the lineage assignment of olfactory neural subtypes between vertebrate species. Identifying the mechanisms underlying the segregation of the various olfactory lineages from overlapping progenitor pools is an important avenue for future research.

## Materials and methods

### Fish husbandry and lines

Fish were maintained at the CBD (Toulouse) and UCI (Irvine) zebrafish facilities in accordance with the rules and protocols in place in the respective locations. The *sox10^{t3}*, *Tg(−2.4gnrh3:egfp)^{zf103}* and *Tg(−4.9sox10:eGFP)^{ba2Tg}* lines have previously been described (*Dutton et al., 2001b*; *Wada et al., 2005*; *Abraham et al., 2008*), as were the *Tg(−28.5Sox10:Cre)^{zf384}* and *Tg(ef1a: loxP-DsRed-loxP-eGFP)^{zf284}* lines used for Cre/loxP-based lineage analyses (*Kague et al., 2012*). Embryos were obtained through natural crosses and staged according to (*Kimmel et al., 1995*).

### In situ hybridisation, immunostaining and microscopy

In situ hybridisation was performed as previously described (*Oxtoby and Jowett, 1993*). Antisense DIG-labelled probes for *gnrh3* (*Abraham et al., 2008*) and *trpc2b* (*Von Niederhäusern et al., 2013*) were generated using standard procedures. In situ hybridisations were visualised using BCIP and NBT (Roche, France) as substrates.

Embryos were immunostained as previously described (*Madelaine et al., 2011*); primary

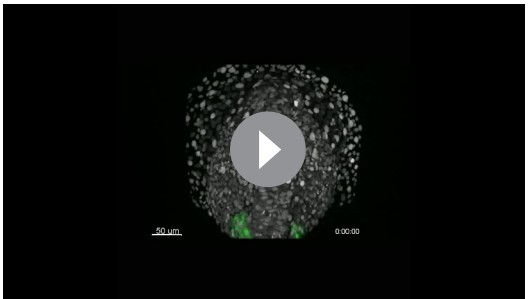

**Video 4.** Microvillous sensory neurons derive from classical preplacodal precursors: tracks. Movie showing an acquisition series and backtracking of sox10:eGFP-negative olfactory cells, lens cells and sox10:eGFP-positive olfactory cells in an Histone2B-RFP loaded *Tg (sox10:eGFP)* transgenic embryo. The movie is divided into 5 parts: an acquisition phase, a backtracking phase for each lineage with tracks and a final representation of the anterior neural plate and adjacent preplacodal ectoderm at 12 hpf showing the combined tracks obtained for the three lineages.

DOI: https://doi.org/10.7554/eLife.32041.018

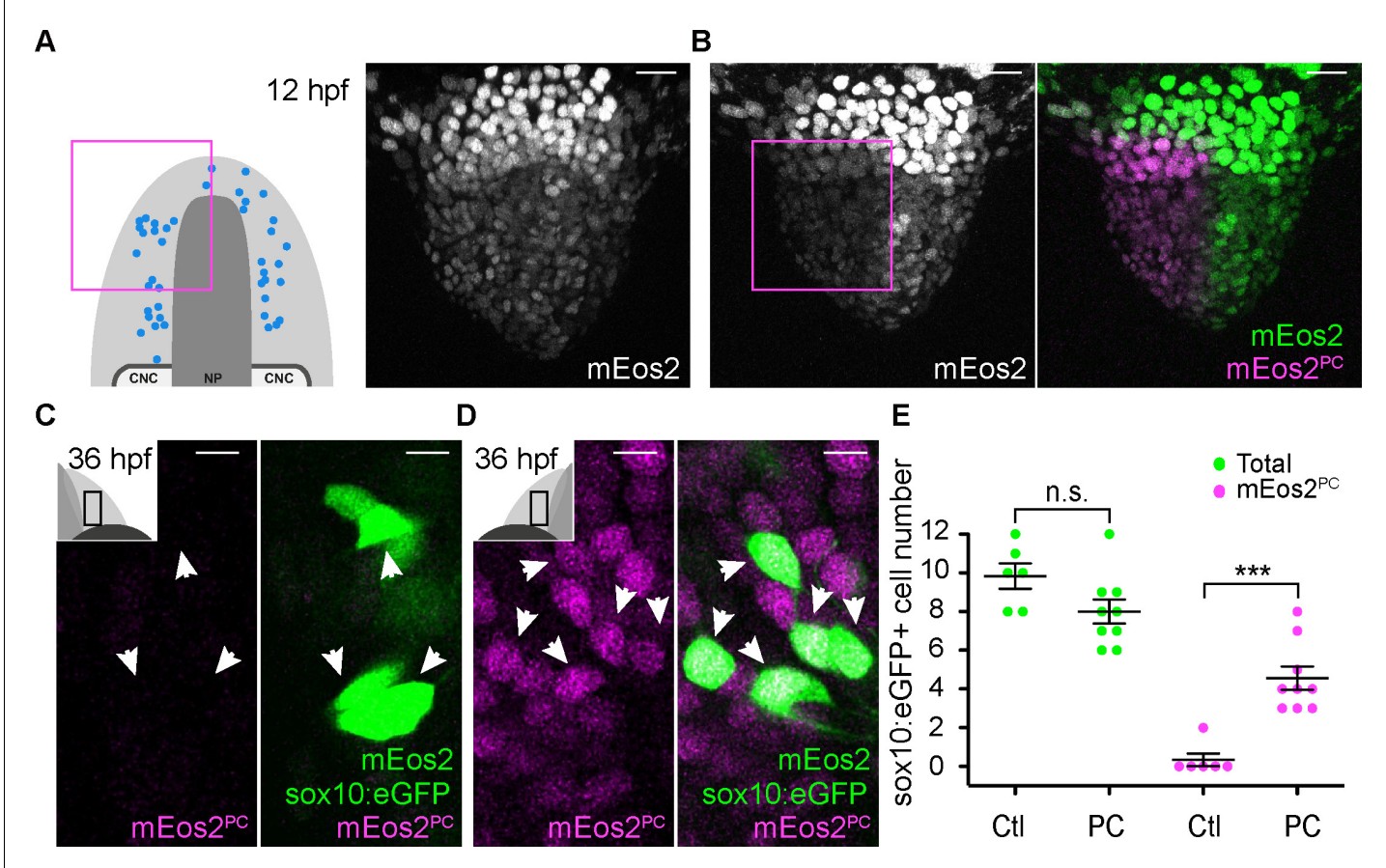

**Figure 6.** Microvillous sensory neurons derive from preplacodal precursors: photoconversion. (A,B) Confocal projections of NLS-mEos2 loaded, *Tg(−4.9sox10:eGFP)* embryos at 12 hpf before (A) and after photoconversion (B). The first panel in (A) shows a schematic representation of the anterior neural plate (NP, dark grey), adjacent preplacodal ectoderm (light grey) and cranial neural crest (CNC) at 12 hpf indicating the origin of backtracked sox10:eGFP-positive cells and the photoconverted area (magenta square), which is also indicated on the projection shown in (B) after photoconversion. Scalebars represent 30 μm. (C,D) Single confocal sections of olfactory epithelia from *Tg(−4.9sox10:eGFP)* embryos at 36 hpf showing the expression of eGFP from the transgene (cytoplasmic green), unconverted NLS-mEos2 (mEos2; nuclear green) and converted NLS-mEos2 (mEos2PC; nuclear magenta). Insets in (C) and (D) shows a schematic representation of an embryonic head at 36 hpf, dorsal view, indicating the area analysed (black rectangle). Scalebars represent 5 μm. (E) Counts of cells expressing eGFP from the *Tg(−4.9sox10:eGFP)* transgene or eGFP and photoconverted mEos2 (mEos2PC) on the control (Ctl) versus photoconverted (PC) sides of the embryo at 36 hpf. No difference in the number of eGFP-positive cells is apparent between the Ctl and PC conditions (9.8 ± 0.7 versus 8 ± 0.6 cells per epithelium, n = 9 and 6, respectively). Conversely, while numerous sox10:eGFP+ cells are also mEos2PC-positive on the photoconverted side, virtually no eGFP/mEos2PC-double positive cells are detected on the control side (4.6 ± 0.6 versus 0.3 ± 0.3 cells per epithelium, n = 9 and 6, respectively). Shown are mean ±s.e.m. P values are calculated using a two-tailed Student's t-test, n.s. not significant, ***p=0.0001.

DOI: https://doi.org/10.7554/eLife.32041.019

The following source data is available for figure 6:

**Source data 1.** sox10:eGFP+ and mEos2PC+ cell number quantification.
DOI: https://doi.org/10.7554/eLife.32041.020

antibodies used were chicken anti-GFP (1:1000; ab13970, Abcam, MA, USA), rabbit anti-GFP (1:1000; TP-401, Torrey Pines Biolabs, CA, USA) and mouse anti-Islet1/2 (1:200; 39.4D5, Developmental Studies Hybridoma Bank, IA, USA). Primary antibodies were detected using the following fluorescently conjugated secondary antibodies (1:1000): Alexa Fluor 680 conjugated donkey anti-mouse IgG (A10038, Molecular Probes, OR, USA), Alexa Fluor 568 conjugated goat anti-mouse IgG (A-11004, Molecular Probes, OR, USA), Alexa Fluor 488 conjugated goat anti-rabbit IgG (A-11034, Molecular Probes, OR, USA) and Alexa Fluor 488 conjugated donkey anti-chicken IgY (703-486-155, Jackson ImmunoResearch, PA, USA). Immunolabellings were counterstained with DAPI (1:1000;

D1306, Life Technologies,CA, USA). Fluorescently labelled embryos were imaged using an inverted Nikon Eclipse Ti Confocal and brightfield images were taken on a Nikon Eclipse 80i microscope. Images were analysed using ImageJ and Imaris 8.3 (Bitplane, Switzerland) software.

### Live confocal imaging and lineages reconstruction

Embryos from the *Tg(gnrh3:egfp)$^{zf103}$* or *Tg(−4.9sox10:eGFP)$^{ba2Tg}$* transgenic lines were injected with synthetic mRNA encoding an H2B-RFP fusion protein. Resulting embryos were grown to 12 hpf, a stage preceding the delamination and anterior migration of cranial neural crest cells (*Schilling and Kimmel, 1994*). They were then dechorionated and embedded for imaging in 0.7% low-melting point agarose in embryos medium in 55 mm round petri dish (Gosselin, France; BP-50). A time-lapse series of confocal stacks (1 µm slice/180 µm deep) was generated of the anterior neural plate and flanking non-neural ectoderm on an upright Leica SP8 Confocal microscope using a 25x HC FLUO-TAR water-immersion objective (L25 × 0.95 W VISIR). Stacks were acquired each 8 min until 36 hpf, a stage when eGFP from either transgene is strongly expressed in the olfactory epithelium. The lineage of the various neuronal populations was subsequently reconstructed manually by backtracking H2B-RFP$^+$ nuclei using Imaris 8.3 analysis software (Bitplane, Switzerland).

### Photoconversion

Embryos from the *Tg(gnrh3:egfp)$^{zf103}$* and *Tg(−4.9sox10:eGFP)$^{ba2Tg}$* transgenic lines were injected with synthetic mRNA encoding an NLS-mEos2 (mEOS2 fused to a nuclear localization sequence) fusion protein (*Sapède et al., 2012*). Embryos were then grown to 12 hpf, dechorionated and embedded for photoconversion/imaging in 0.7% low-melting point agarose in embryos medium in 35 mm circular petri dish (Nunc; 153066) bearing a silicone sealed 22 mm circular cover slip (Thermo Scientific, France; 174977). Mounted embryos were first imaged for NLS-mEos2 expression prior photoconversion at very low laser levels (confocal stack 2 µm slice/80 µm deep). Subsequently, a region of interest (ROI) was photoconverted using a 405 nm diode (100% laser, 41 s), after which embryos were imaged again to assess the extent of NLS-mEos2 conversion. Photoconversion and imaging was done on an inverted SP8 Leica confocal with an HC PL APO CS2 40x/1.3 oil objective. Full z-stacks were acquired for each photoconverted embryo 24 hr after the photoconversion (confocal stack 1 µm slice/80 µm deep) to determine the contribution of progenitors located in the ROI at the time of photoconversion to the Gnrh3 or microvillous lineages.

### Statistical analysis

All statistical comparisons are indicated in figure legends including one sample and unpaired t-test performed using Prism (GraphPad.) The scatter dot plots were generated with Prism. Data are mean ± s.e.m. Two-tailed t-test *p<0.05, **p<0.01, ***p<0.005, ****p<0.0001.

## Acknowledgements

This work was supported by the Centre National de la Recherche Scientifique (CNRS); the Institut National de la Santé et de la Recherche Médicale (INSERM); Université de Toulouse III (UPS); University of California, Irvine; National Institute of Health (NIH R01 DE13828 and AR67797 to TS); Fondation pour la Recherche Médicale (DEQ20131029166 to PB); Fédération pour la Recherche sur le Cerveau; and the French Ministère de la Recherche. We would like to thank Brice Ronsin and the Toulouse RIO Imaging platform (LITC), and Aurore Laire, Richard Brimicombe-Lefevre and Ines Gehring for fish husbandry. We thank the Alsina, Fisher, Granato, Kelsh, Neuhauss, and Zohar labs for providing fish lines and other reagents. We also thank Christian Mosimann, Tatjana Sauka-Spengler and Trevor Williams for comments and suggestions.

## Additional information

### Funding

| Funder | Grant reference number | Author |
|---|---|---|
| Fondation pour la Recherche Médicale | DEQ20131029166 | Raphaël Aguillon<br>Julie Batut<br>Romain Madelaine<br>Pascale Dufourcq<br>Patrick Blader |
| National Institute of Dental and Craniofacial Research | DE13828 | Arul Subramanian<br>Thomas F Schilling |
| National Institute of Arthritis and Musculoskeletal and Skin Diseases | AR67797 | Arul Subramanian<br>Thomas F Schilling |
| National Institutes of Health | R01 DE13828 | Thomas F Schilling |
| Fédération pour la Recherche sur le Cerveau | | Patrick Blader |

The funders had no role in study design, data collection and interpretation, or the decision to submit the work for publication.

### Author contributions

Raphaël Aguillon, Patrick Blader, Conceptualization, Supervision, Funding acquisition, Investigation, Writing—original draft, Writing—review and editing; Julie Batut, Pascale Dufourcq, Conceptualization, Investigation, Writing—original draft, Writing—review and editing; Arul Subramanian, Conceptualization, Supervision, Investigation, Writing—original draft, Writing—review and editing; Romain Madelaine, Investigation; Thomas F Schilling, Investigation, Writing—original draft, Writing—review and editing

### Author ORCIDs

Julie Batut (iD) https://orcid.org/0000-0002-1984-2094
Arul Subramanian (iD) http://orcid.org/0000-0001-8455-6804
Thomas F Schilling (iD) http://orcid.org/0000-0003-1798-8695
Patrick Blader (iD) http://orcid.org/0000-0003-3299-6108

### Ethics

Animal experimentation: Animal experimentation: The study was performed in strict accordance with French and European guidelines (Toulouse), and to the recommendations in the Guide for the Care and Use of Laboratory Animals of the National Institutes of Health (Irvine). Toulouse, France: French veterinary service and national ethical committee approved the protocols in this study, with approval ID: A-31-555-01 and APAPHIS #3653-2016011512005922v6. Irvine, USA: All of the animals were handled according to approved institutional animal care and use committee (IACUC) protocols (#2000-2149) of the University of California, Irvine. The renewal of this protocol was approved by the IACUC (Animal Welfare Assurance #A3416.01) on December 11, 2015. All animal experiments were performed on embryos derived from natural spawnings and every effort was made to minimize suffering.

### Decision letter and Author response

Decision letter https://doi.org/10.7554/eLife.32041.023
Author response https://doi.org/10.7554/eLife.32041.024

## Additional files

**Supplementary files**

• Transparent reporting form
DOI: https://doi.org/10.7554/eLife.32041.021

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
