## [Decision Letter]

Thank you for submitting your article "Cell-type heterogeneity in the zebrafish olfactory placode is generated from progenitors within preplacodal ectoderm" for consideration by *eLife*. Your article has been reviewed by three peer reviewers, and the evaluation has been overseen by Tanya Whitfield as the Reviewing Editor and Didier Stainier as the Senior Editor. The following individual involved in review of your submission has agreed to reveal his identity: Joachim Wittbrodt (Reviewer #1).

The reviewers have discussed the reviews with one another and the Reviewing Editor has drafted this decision to help you prepare a revised submission.

Summary:

The developmental origin of GnRH neurons within the olfactory placode has been debated in the literature as it is often not clear whether the lines used for genetic lineage tracing of the neural crest are entirely specific. The manuscript entitled "Cell-type heterogeneity in the zebrafish olfactory placode is generated from progenitors within preplacodal ectoderm" by Aguillon and colleagues addresses the origin of different neuronal cell types with the olfactory placode and finds, in contrast to the literature, no evidence for contribution of the cranial neural crest to either GnRH or microvillous sensory neurons. Using classical approaches in a modern interpretation (live imaging and cell tracking), the authors establish a precise 4D fate map for the olfactory placode. This, in combination with cell type-specific markers, allows them to trace cells back to the preplacodal ectoderm. To address the contribution of the cranial neural crest they study strong Sox10 mutant alleles that fail to properly specify migrating cranial crest. The authors propose a model in which the cell type heterogeneity correlates with the initial position within the preplacodal ectoderm.

The study combines four different approaches – analysis of sox10 mutants; Cre/lox lineage-tracing of Sox10-expressing cells [using *Tg(-28.5Sox10:Cre);Tg(ef1a:loxP-DsRed-loxP-eGFP*) embryos]; manual lineage reconstruction from time-lapse confocal movies of *Tg(gnrh3:eGFP)* embryos, plus unilateral photoconversion of the Gnrh3 precursor domain (identified from the backtracking experiments) in *Tg(gnrh3:eGFP)* embryos – to show convincingly that GnRH neurons in zebrafish arise from preplacodal ectoderm, not the neural crest as previously proposed (from DiI-labeling the CNC and *sox10* morphant experiments).

The authors similarly use *sox10* mutant analysis, backtracking and photoconversion in *Tg(-4.9sox10:eGFP)* embryos – in which GFP expression was previously reported to overlap almost completely with the microvillous sensory neuron reporter *Tg(-4.5trpc2b:GAP-Venus)* – to show that microvillous sensory neurons arise from preplacodal ectoderm, not the neural crest.

The manuscript is well crafted and presents data of excellent quality. The analyses are carefully controlled and supported by extensive supplementary material.

The videos provided illustrate the process of the fate mapping approach and are clear and instructive as well. The data are convincing and well presented, with multiple lines of evidence supporting the conclusions drawn, which resolve a longstanding controversy. The findings are consistent with more recent studies in the chick and mouse and suggest an evolutionarily conserved lineage trajectory for these preplacodal progenitors across vertebrates. Appropriate consideration for previous studies is given and this study additionally acts as a caution for the interpretation of morphant phenotypes and the use of transgenics in lineage tracing studies. Overall, the authors present a clear and convincing case.

Essential revisions:

1) Given that the data contradict a model in the literature based on morpholino data, this should be discussed a bit more extensively. Does the mutant still exhibit its original phenotype?

2) The second point is that even though the authors perform a 4D analysis, this is unfortunately not presented in the study where they provide starting and end point analyses. The audience would profit from a presentation of migratory tracks of the cell types followed.

3) The GFP signal in Figure 2, Figure 3 and Figure 3—figure supplement 1 (in the latter two Figures, at 36h) is somewhat obscured even by the DAPI: please show the GFP channel alone, as well, to allow direct comparison with the dual/triple-channel images.

4) For Figure 2, given the importance of this result, it would be helpful to show higher-power views of the Islet-positive cells (as in e.g. Figure 4), and additional image(s) of the same embryo showing the "widespread expression of eGFP throughout the head" reported in the text, i.e., GFP expression in expected (and labeled) neural crest derivatives.

---

## [Author Response]

Essential revisions:1) Given that the data contradict a model in the literature based on morpholino data, this should be discussed a bit more extensively. Does the mutant still exhibit its original phenotype?

We have added a new paragraph to the Discussion dealing with the disparity between mutant and morphant phenotypes. The *sox10^t3^* allele used in our studies reproduces the strong pigment and ear phenotypes previously reported by Dutton and colleagues, and this is mentioned in the new paragraph.

2) The second point is that even though the authors perform a 4D analysis, this is unfortunately not presented in the study where they provide starting and end point analyses. The audience would profit from a presentation of migratory tracks of the cell types followed.

We initially wondered about putting tracks in Figure 3 and Figure 5 but attempts to do so did not provide satisfying results. This predominantly stems from the fact that 1) the figures were already "complex" and adding 25-30 tracks obscured the data rather than enhancing it and 2) we were unsure whether the tracks would be better overlaid on the 36hpf or 12hpf confocal projections. To address the reviewers' request, we have chosen to include a pair of supplemental videos (new Video 2 and Video 4). These contain the data of the single embryo presented in Figure 3 and Figure 5, respectively, and the tracks for all cells analysed in these embryos. This has the advantage of not obscuring the "starting versus end point" information in the original figures, while also providing an appreciation of the dynamic cell movements.

3) The GFP signal in Figure 2, Figure 3 and Figure 3—figure supplement 1 (in the latter two Figures, at 36h) is somewhat obscured even by the DAPI: please show the GFP channel alone, as well, to allow direct comparison with the dual/triple-channel images.

As requested, the figures mentioned have been modified to include the GFP channel alone.

4) For Figure 2, given the importance of this result, it would be helpful to show higher-power views of the Islet-positive cells (as in e.g. Figure 4), and additional image(s) of the same embryo showing the "widespread expression of eGFP throughout the head" reported in the text, i.e., GFP expression in expected (and labeled) neural crest derivatives.

Along with a panel showing the GFP channel alone, we have included a panel showing a higher magnification of the Islet-positive cells. However, rather than adding panels highlighting the extent to which eGFP can be found throughout the head of double-heterozygous *Tg(-28.5Sox10:Cre);Tg(ef1a:loxP-DsRed-loxP-egfp)* embryos at 48 hpf, we have provided this data in Figure 2—figure supplement 1. The data used to construct the figure supplement was taken from a confocal dataset that we had already acquired for the embryo shown in the original Figure 2.